

# Factors encouraging mobile instant messaging service use in medical education

Kidong Kim[1], Banghyun Lee[2], Youngmi Park[3], Eun Young Jung[4], Seul Ki Kim[1], Dong Hoon Suh[1] and Bo Ram Choi[2]

[1] Department of Obstetrics and Gynecology, Seoul National University Bundang Hospital, Seongnam-Si, Gyeonggi-Do, South Korea
[2] Department of Obstetrics and Gynecology, Hallym University, Kangdong Sacred Heart Hospital, Seoul, South Korea
[3] Division of Statistics, Medical Research Collaborating Center, Seoul National University Bundang Hospital, Seongnam-Si, Gyeonggi-Do, South Korea
[4] Win Women's Hospital, Suwon-Si, Gyeonggi-Do, South Korea

## ABSTRACT

**Background**. Mobile instant messaging services are being increasingly used for educational purposes, but their effectiveness in medical education is not well known. We assessed whether students' use of Kakao Talk (a mobile instant messaging service) during the early period of a week of clinical education influenced its use for academic purposes during a later period of the same week.

**Methods**. The online communication records of 151 third-year medical students (in 39 clinical education groups) who used Kakao Talk during clinical education were reviewed. The 39 groups were categorized as low, middle, or high according to the number of total chats (on all subjects, not just academic) per student over five days. The relationship between the number of total chats during the first two days and the number of academic chats during the last three days (of five-day chatroom weeks) was analyzed.

**Results**. The number of total and academic chats over all five days, the first two days, and the last three days was highest in groups with the highest number of total chats per student. Similarly, the highest number of students posting total and academic chats was found in these groups. In addition, the number of academic chats per student and the frequency of questions raised by students were also highest in these groups. During the last three days, the number of students posting total chats was lower than that during the first two days, and the number of academic chats per student posting academic chats was higher. The number of total chats on the first or second day positively correlated with the maximum value of academic chats on the third to fifth days.

**Conclusion**. High frequency mobile instant messaging use early on in clinical education might encourage its use for academic purposes during later periods.

Corresponding author
Banghyun Lee,
banghyun.lee@gmail.com

## INTRODUCTION

Worldwide, the use of mobile instant messaging (MIM) services is rapidly increasing. Their relative convenience, accessibility, and capacity to disseminate a large amount of information in various formats suggest that MIM services have high potential for educational use. Thus, the use of this service in educational contexts seems inevitable. In many studies, the influence of MIM use on students' academic performances has been shown to be positive, to be negative, or to have no effect, regardless of factors such as the level of students' education (*Alipour et al., 2012*; *Lauricella & Kay, 2013*; *Bouhnik & Deshen, 2014*; *Yeboah & Ewur, 2014*; *Jamal et al., 2016*; *So, 2016*; *Goyal, Tanveer & Sharma, 2017*; *Raiman, Antbring & Mahmood, 2017*; *Clavier et al., 2019*). Several studies have reported that university students and faculties have positive perceptions of MIM services and accept their use in teaching and learning (*Lauricella & Kay, 2013*; *Bouhnik & Deshen, 2014*; *Yeboah & Ewur, 2014*; *So, 2016*).

In the field of medical education, a few small studies have reported that MIM might be an effective way to acquire medical knowledge and a useful, feasible, and acceptable tool for medical students and residents (*Alipour et al., 2012*; *Jamal et al., 2016*; *Goyal, Tanveer & Sharma, 2017*; *Raiman, Antbring & Mahmood, 2017*). However, other studies showed MIM services to be ineffective tools in educating medical residents, suggesting instead that they served as a distraction (*Clavier et al., 2019*). Moreover, social media such as WhatsApp Messenger (an MIM service, WhatsApp Inc., Mountain View, California, USA), YouTube (YouTube, LLC., San Bruno, California, USA), and Twitter (Twitter, Inc., San Francisco, California, USA) showed no influence on the academic performance of medical students (*AlFaris et al., 2018*). Given the growing use of MIM services in education, it is necessary to clarify their possible roles in medical education.

Kakao Talk (aka KaTalk, Kakao Corp, Seoul, Korea) is a user-friendly MIM service utilized by about 93% of smartphone owners of all ages in South Korea (*Nay, 2013*). This free MIM service, which is available in 15 languages, includes free text messaging and free voice calls (*Han & Cho, 2015*; *Unuth, 2019*). One might expect that Kakao Talk would be useful in medical education because of its ubiquity in South Korea for interpersonal and person-to-group communication as well as for information sharing.

This study intended to determine the efficacy of the use of Kakao Talk by senior medical students during their clinical education. On the basis of its ubiquity, accessibility, and convenience, we assumed that Kakao Talk would be frequently used for academic purposes when it is seen as being helpful. We further hypothesized that an increase in student activity on Kakao Talk would be related to its later use for academic purposes. This is because it is likely that students who want to improve their clinical education would post more academically related chats once they become familiar with a learning environment that uses MIM.

## MATERIALS AND METHODS

### Sample

This retrospective and exploratory pilot study used data on senior medical students in the clinical phase of their education at the Seoul National University Bundang Hospital. Data were collected between November 10, 2014 and December 5, 2015. The Institutional Review Board of Seoul National University Bundang Hospital approved the study design (No. L-2016-1265) on November 15, 2016. Informed consent was waived.

Participants were third-year medical students ($n = 151$) receiving clinical education in the Department of Obstetrics and Gynecology. Moreover, professors ($n = 3$; two oncologics and one endocrinologist) and fellows ($n = 2$; one oncologist and one obstetrician) majoring in obstetrics and gynecology as well as residents ($n = 12$) in that department also participated in the use of Kakao Talk for educational purposes.

### Study design

Third-year medical students at this institution receive clinical education in the Department of Obstetrics and Gynecology (a one-week rotation, Monday through Friday) in groups of three to five students. Each student is assigned to an academic advisor who is a professor of obstetrics and gynecology. The students attend in the delivery room, operating room, infertility intervention room, wards, and outpatient clinic. On Thursdays, they attend an obstetrics and gynecology course. Additionally, on Monday of the rotation week, each student is assigned the medical case of a patient admitted to the obstetrics and gynecology ward. He or she presents that case at a conference held on Friday of the same week (Fig. 1).

We posited that using a virtual space such as a chat room might be a valuable supplement to formal education during clinical education. To this end, Kakao Talk was introduced as an informal educational tool in November 2014. On Monday mornings, the hospital resident responsible for education created a chat room on Kakao Talk and invited students and teachers to participate. At this time, students were informed that neither the extent nor the content of their activities on Kakao Talk would influence their clinical scores/grades. During the week, students could post questions in the chatroom. Teachers (either professors or fellows) responded to each question as quickly as possible. The chat rooms were then closed on Saturday morning of that week (Fig. 1).

### Research questions

Over the week-long clinical instruction period, we observed (without any quantification) that unfamiliarity among students and between students and teachers on Kakao Talk during the early part of a week induced fewer chat postings during that week. Therefore, we hypothesized that the sooner students began using Kakao Talk, the more they might use it overall and the more they might use it for academic purposes (i.e., chat postings related to medicine, especially obstetrics and gynecology, rather than social content). In this study, the relationship between the total number of student chats during the first two days of the week and the number of academic chats (defined as chats related to medicine) during the last three days was evaluated as the primary outcome. Moreover, the frequency with which

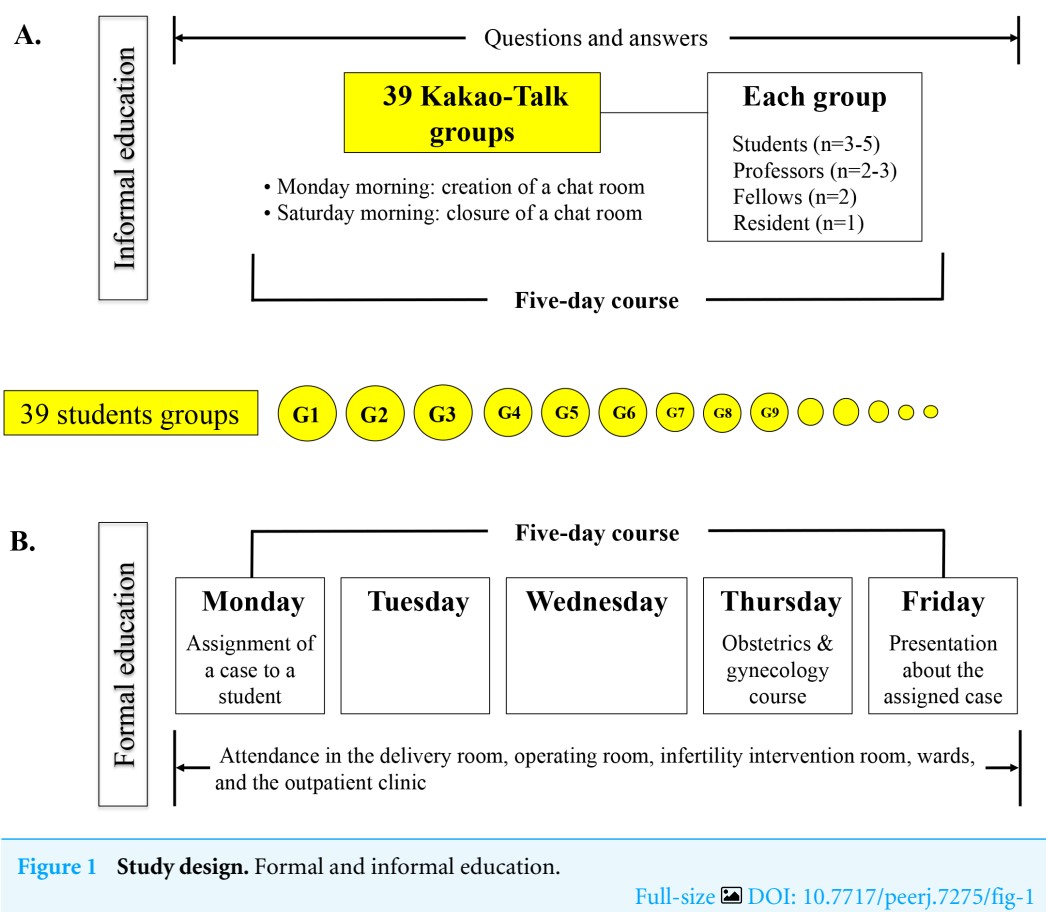

**Figure 1  Study design.** Formal and informal education.

students used Kakao Talk during the clinical education period and the factors associated with the frequency of this use were examined.

## Data collection and analysis

Thirty-nine weeks of online communication records of students' educational use of Kakao Talk were reviewed. A total of 39 clinical education groups ($n = 151$) were categorized into three sets (Low, $n = 13$; Middle, $n = 13$; High, $n = 13$) according to the number of total chats per student over five days. Next, the three categories (Category 1 = Low, Category 2 = Middle, Category 3 = High) were determined according to the number of chats, sex ratio, season of clinical education (spring, summer, fall, winter), number of student questions, type of student questions (according to the subspecialty of obstetrics and gynecology), response time to student questions, and number of students (Tables 1 and 2).

We used Poisson regression to estimate the scatter plot and correlation coefficient regarding the relationship between the number of total chats during the first and second days and the maximum value of academic chats during the third through fifth days. Categorical variables were expressed as counts and percentages and analyzed using Fisher's exact test. For data with normal distributions, continuous variables were expressed using the means ± standard deviations, while medians and interquartile ranges were used for data with non-normal distributions.

**Table 1 Characteristics based on the number of total chats per student.**

| Characteristic | Total (n = 39) | Category | | | p-value |
|---|---|---|---|---|---|
| | | 1 (n = 13) | 2 (n = 13) | 3 (n = 13) | |
| Number of total chats per student, median (interquartile range) | 2.3 (2) | 1.3 (0.6) | 2.3 (0.7) | 6.5 (5.4) | 0.000 [a,b,c] |
| Number of students, n (%) | | | | | 0.416[*] |
| Three | 8 (20.5) | 1 (2.5) | 4 (10.3) | 3 (7.7) | |
| Four | 29 (74.4) | 12 (30.8) | 8 (20.5) | 9 (23.1) | |
| Five | 2 (5.1) | 0 (0) | 1 (2.5) | 1 (2.5) | |
| Sex ratio of students, n (%) | | | | | 0.437[*] |
| Equal (2:2) | 8 (20.5) | 2 (5.1) | 1 (2.5) | 5 (12.8) | |
| Male > female | 26 (66.7) | 9 (23.1) | 10 (25.6) | 7 (17.9) | |
| Male < female | 5 (12.8) | 2 (5.1) | 2 (5.1) | 1 (2.5) | |
| Season, n (%) | | | | | 0.278[*] |
| Spring | 12 (30.8) | 4 (10.3) | 2 (5.1) | 6 (15.4) | |
| Summer | 8 (20.5) | 1 (2.5) | 3 (7.7) | 4 (10.3) | |
| Fall | 16 (41.0) | 6 (15.4) | 7 (17.9) | 3 (7.7) | |
| Winter | 3 (7.7) | 2 (5.1) | 1 (2.5) | 0 (0) | |

**Notes.**

p-value for the difference among the categories by multiple comparisons using the Kruskal–Wallis test, followed by the 2-tailed Mann–Whitney rank sum test.

Differences are reported between

[a] category 1 v. category 2.

[b] category 2 v. category 3.

[c] category 1 v. category 3.

*p- value for differences among the categories using Fisher's exact test.

'Number of total chats per student' was defined as the total number of chats divided by the total number of students in each group.

Differences among groups were analyzed using one-way analysis of variance (ANOVA) followed by Bonferroni's method for parametric data and using the Kruskal–Wallis test followed by the two-tailed Mann–Whitney rank sum test for nonparametric data. The cut-off used to determine statistical significance was $p < .05$. We used the open-source statistical software R version 3.3.0 (http://www.R-project.org) and SPSS version 23 (IBM Corporation, Inc., Chicago, IL, USA) for the computations.

## RESULTS

The characteristics of the 39 groups showed that there were more four-student groups, more groups with more males than females, and fewer winter groups than other types of groups. Distribution according to the number of students and sex ratio of students in the clinical groups and according to season did not differ across the three categories (Table 1). Among participants in the third category (high number of chats), the following were all found to be significantly higher than in the other two categories: (1) the number of total and academic chats over all five days, the first two days, and the last three days, (2) the number of students posting total and academic chats, (3) the number of academic chats per student, (4) the number of student questions, and (5) the number of questions related to obstetrics, endocrinology, gynecology, and gynecological oncology (Table 2).

**Table 2  Student activity on Kakao Talk based on the number of total chats per student.**

| Student chat activity | Total (n = 39) | Category | | | p-value |
|---|---|---|---|---|---|
| | | 1 (n = 13) | 2 (n = 13) | 3 (n = 13) | |
| Student chats | | | | | |
|   Total, mean ± SD | 12.7 ± 10.8 | 5.2 ± 1.7 | 8.9 ± 2.0 | 24 ± 12.1 | .000 [b,c] |
|   Academic, mean ± SD | 9.8 ± 11.3 | 3.5 ± 3.2 | 4.5 ± 2.7 | 21.5 ± 12.9 | .000 [b,c] |
| Student chats during first and second days | | | | | |
|   Total, mean ± SD | 6 ± 5.4 | 3.2 ± 1.9 | 5.2 ± 2.4 | 9.6 ± 7.8 | .006 [b,c] |
|   Academic, median (interquartile range) | 1 (6) | 0 (3.5) | 0 (5) | 6 (13.5) | .056 [b,c,*] |
| Student chats during the last three days, median (interquartile range) | | | | | |
|   Total | 5 (10) | 2 (2.5) | 4 (3.5) | 13 (10) | .000 [a,b,c,*] |
|   Academic | 3 (8) | 1 (2) | 2 (4.5) | 13 (11) | .000 [b,c,*] |
| Number of students posting chats, median (interquartile range) | | | | | |
|   Total | 4 (1) | 3 (1.5) | 4 (1) | 4 (0) | .003 [a,c,*] |
|   Academic | 3 (2) | 1 (2) | 2 (1) | 4 (0.5) | .000 [b,c,*] |
| Number of academic chats per student, median (interquartile range) | 1.4 (2.2) | 0.5 (1.1) | 1.0 (1.2) | 5.8 (5.3) | .000 [b,c,*] |
| Number of posted questions, median (interquartile range) | 3 (6) | 1 (2.5) | 3 (2) | 9 (11) | .000 [a,b,c,*] |
| Types of posted questions, median (interquartile range) | | | | | |
|   Obstetrics | 1 (2) | 0 (1) | 1 (1) | 3 (4) | .001 [b,c,*] |
|   Endocrinology | 0 (1) | 0 (1) | 0 (1) | 2 (3) | .000 [b,c,*] |
|   Gynecology[d] | 0 (1) | 0 (1) | 1 (2) | 2 (3) | .015 [a,c,*] |
|   Gynecologic oncology | 1 (2) | 0 (1) | 0 (1) | 3 (3) | .002 [b,c,*] |

**Notes.**

  p- value for differences among the categories by multiple comparisons using one-way ANOVA, followed by the Bonferroni method.

  *p-value for the difference among the categories by multiple comparisons using the Kruskal–Wallis test, followed by the 2-tailed Mann–Whitney rank sum test.

  Differences are reported between

[a] category 1 v. category 2, in total, student chats during the last three days was 0.064.

[b] category 2 v. category 3, in total and academic student chats during first and second days were 0.071 and 0.081, respectively.

[c] category 1 v. category 3.

[d] Gynecology included urogynecology.

  'Number of total chats per student' was defined as the total number of chats divided by the total number of students in each group.

  'Number of academic chats per student' was defined as the number of academic chats divided by the total number of students in each group.

The mean response time to student questions and the proportion of student questions to which teachers replied within 30 min to one hour of posting were not significantly different across categories ($p > .05$; File S1).

On the first day of the clinical education week, 82.1% of groups posted total chats, while on the second day, the percentage was 46.2%, on the third day, 61.5%, on the fourth day, 51.3%, and on the fifth day, 41.0%. The proportions of students posting total chats were 66.9% on the first day, 28.5% on the second day, 36.4% on the third day, 29.8% on the fourth day, and 25.2% on the fifth day. More total chats than academic chats were posted on the first and fifth days ($p = .001$ and $p = .035$, respectively). On the second, third, and fourth days, the number of total chats was similar to the number of academic chats ($p > .05$; File S1). The number of total and academic chats, the number of students posting academic

**Table 3 Student activity on Kakao Talk during the first two days and the last three days of a clinical week.**

| Student activity | Total ($n = 39$) | First two days ($n = 39$) | Last three days ($n = 39$) | p-value |
|---|---|---|---|---|
| Student chats, median (interquartile range) | | | | |
|     Total | 8 (8) | 5 (4) | 5 (10) | .756 |
|     Academic | 6 (8) | 1 (6) | 3 (8) | .105 |
| Number of students posting chats, median (interquartile range) | | | | |
|     Total | 4 (1) | 3 (2) | 2 (3) | .038 |
|     Academic | 3 (2) | 1 (3) | 2 (3) | .243 |
| Number of chats per student posting chats, median (interquartile range) | | | | |
|     Total | 3.5 (1.25) | 1.5 (1) | 2 (1.75) | .177 |
|     Academic | 2.5 (2) | 1 (2) | 2 (2.75) | .032 |

**Notes.**
$p$- value for the difference between the first two days and the last three days using the 2-tailed Mann–Whitney rank sum test.
'Number of total chats per student posting chats' was defined as the total number of chats divided by the total number of students posting chats in each group.
'Number of academic chats per student posting chats' was defined as the number of academic chats divided by the number of students posting academic chats in each group.

chats, and the number of total chats per student posting total chats were not significantly different between the first two days and the last three days. However, during the last three days, the number of students posting total chats was lower than in the first two days, and the number of academic chats per student posting academic chats was higher (Table 3). The number of total and academic chats per student was not significantly different between the first two days and the last three days (total, median [interquartile range]: 1.3 [1.3] vs 1.3 [2.1], $p = .787$; academic: 0.3 [1.5] vs 1 [2], $p = .101$). Because of the extremely small number of total and academic chats per student, the number of chats per student posting chats was evaluated.

To test whether the number of total chats during the first two days influenced the number of academic chats during the last three days, data on the groups in which the students posted chats during the first two days were analyzed. As shown in Fig. 2A, the number of total chats posted on the first day significantly and positively correlated with the maximum value of academic chats on the third through fifth days (first day: correlation coefficient = 0.43, $p = .02$). However, those posted on the second day (Fig. 2B) showed a positive correlation with no statistical significance (second day: correlation coefficient = 0.37, $p = .17$).

In the third category, all students in all groups (median four students) posted total chats as well as academic chats. Meanwhile, in the second category, all students in all groups except two posted total chats (median four students), while some students in all except one group (median two students) posted academic chats. By contrast, in the first category, some students in all except three groups posted total chats (median three students), and some students in all groups except three (median one student) posted academic chats. We found that the number of total and academic chats was higher during the last three days than during the first two days in 11 groups in the third category (the number was equal in one group and lower in one group). This was also true of the second category in six groups (lower in seven groups), while in the first category, this was true for three

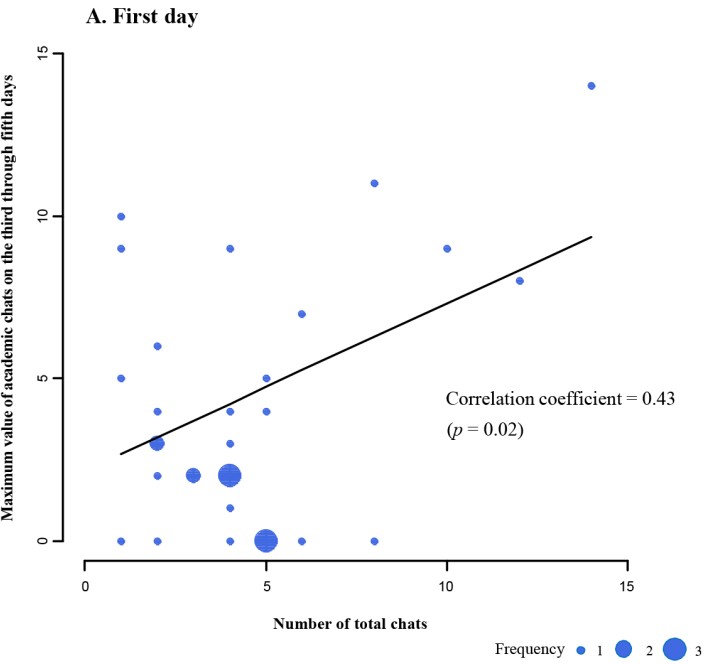

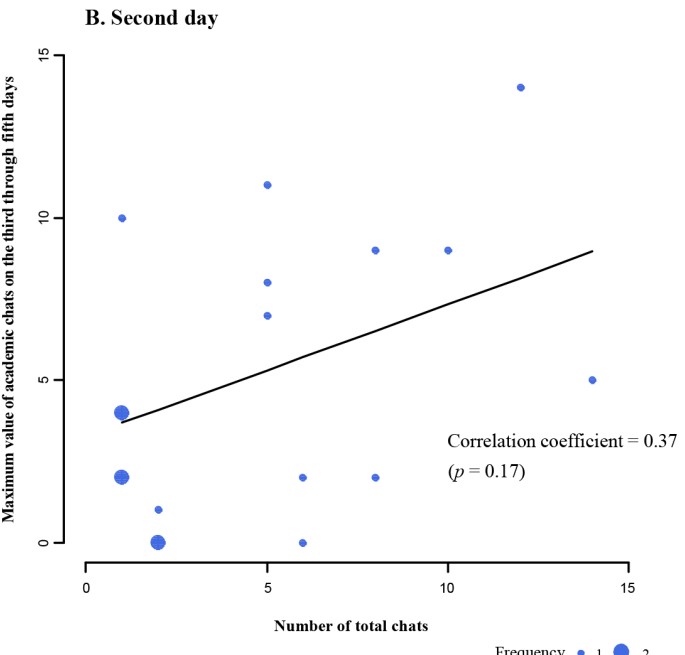

**Figure 2   Scatter plots of maximum value of academic chats on the third through fifth days by the number of total chats on the first or second day.** (A) First day and (B) Second day. Frequency refers to the number of groups performing the same number of chats.

groups (equal in one group and lower in nine groups). In all categories, the subject most frequently discussed in academic chats was the medical case allocated to each student, as well as procedures and delivery observed during clinical education and obstetrics and gynecology courses attended (File S1).

## DISCUSSION

In this exploratory pilot study, we retrospectively reviewed 39 weeks of online communication by 151 senior medical students related to their use of the Kakao Talk MIM service as an informal modality for education. The frequency with which the students used Kakao Talk during the first two days of a week of clinical education positively correlated with the frequency with which they used it for academic purposes during the last three days of that week. The number of students posting total chats decreased and the number of academic chats per student posting academic chats increased over one week of clinical education. In addition, the number of total chats per student positively correlated with the following: (1) the number of total and academic chats over all five days, the first two days, and the last three days, (2) number of students posting total and academic chats, (3) the number of academic chats per student, and (4) with the number and types of student questions.

Results from previous studies concerning the effectiveness of the use of MIM services in medical education vary. One prospective study on improving knowledge on breast cancer among 25 hospital residents of gynecology and obstetrics reported that MIM services had a more positive influence on learning and created more interest among learners than did booklets (Alipour et al., 2012). In another prospective study, WhatsApp successfully facilitated a variety of educational interactions in six groups of 19 third-year medical students (with one tutor per group) during an eight-week clinical education period (Raiman, Antbring & Mahmood, 2017). In a pilot study, WhatsApp was found to be useful in training 45 pathology residents (Goyal, Tanveer & Sharma, 2017). However, in a randomized, multicenter small study, the use of WhatsApp ($n = 32$) was less effective for teaching medical residents than traditional e-learning ($n = 30$), and it was suggested that its use served as a distraction (Clavier et al., 2019). In addition, a recent study ($n = 400$) reported that while almost all medical students (98% of the 97% who responded to the questionnaire) in the study used social media (WhatsApp, YouTube, and Twitter), only a minority of students (40%) used such platforms for academic purposes, suggesting that the use of social media has no influence on academic performance (AlFaris et al., 2018). In the current study, more than 25% of students continually used Kakao Talk during the study period. During the middle of the clinical education week (second through fourth days), students mainly used Kakao Talk for academic purposes. The number of students posting chats and the number of chats per student who posted chats showed that Kakao Talk was more frequently used for academic than for social purposes over a one-week period of clinical education. Moreover, students who used Kakao Talk most often during the early period also used it more often over time for academic purposes. These results demonstrate that using a MIM service, such as Kakao Talk, might be useful in medical
students' education but that the timing of the start of this communication might also be a key factor in its effectiveness.

In this study, the frequency with which students made use of virtual space on Kakao Talk during clinical education was not influenced by teachers' response times to student questions. The students had different levels of activity on Kakao Talk, and many students used it for academic purposes. The use of a MIM service by some, but not all, students might be due to their being unacquainted with one another and with their teachers on Kakao Talk during the early part of the clinical education period, which in turn, might influence their preference for MIM service use for academic purposes. Our results show a positive relationship between total activity on Kakao Talk during the early period and the number of academic chats during the later period, indicating that early chatting among students and teachers increases online chat activity overall followed by increased use for academic purposes.

The number of total and academic chats per student can be viewed as representing the extent of students' voluntary usage of Kakao Talk. Our data demonstrate that in the groups categorized into the third category, teamwork was stronger than in groups in other categories. This is because all students participated in posting total chats, almost all students participated in posting academic chats, and the number of total and academic chats increased over time in most of these groups. Students in groups in the first category showed relatively weaker teamwork based on their low (individual) participation in posting chats (total and academic) and the decrease in the number of chats (total and academic) over time in many groups. Students with a stronger sense of teamwork might use Kakao Talk in an environment where they are encouraged to use this tool for social and academic purposes. This, in turn, might lead to relatively higher academic exposure for all students. In contrast, students with a weaker sense of teamwork might tend to use Kakao Talk individually in an environment where its use is not encouraged, which may lead to relatively lower academic exposure for all students.

The limitations of our study include its retrospective nature and relatively short observation period. Moreover, the efficacy of MIM services in student education was not directly evaluated in relation to clinical performance, test scores, or other forms of assessment. Nevertheless, by suggesting that the use of a popular MIM service might be related to an eventual increase in its use for academic purposes within a very short period, this pilot study has potential as a basis for well-designed large-scale prospective trials.

## CONCLUSION

This pilot study demonstrated that if MIM services are used frequently by medical students during the early part of a period of clinical education, they are more likely to be used for academic purposes later on. Our study also suggests that MIM services (particularly Kakao Talk) can be used for medical students' education, although it is an informal modality. Large-scale prospective trials are warranted to further evaluate the utility of MIM services.

### Funding

The authors received no funding for this work.

### Competing Interests

The authors declare there are no competing interests.

### Author Contributions

- Kidong Kim conceived and designed the experiments, performed the experiments, analyzed the data, contributed reagents/materials/analysis tools, prepared figures and/or tables, authored or reviewed drafts of the paper, approved the final draft.
- Banghyun Lee conceived and designed the experiments, performed the experiments, contributed reagents/materials/analysis tools, prepared figures and/or tables, authored or reviewed drafts of the paper, approved the final draft.
- Youngmi Park analyzed the data, contributed reagents/materials/analysis tools, authored or reviewed drafts of the paper, approved the final draft.
- Eun Young Jung, Seul Ki Kim and Dong Hoon Suh conceived and designed the experiments, performed the experiments, authored or reviewed drafts of the paper, approved the final draft.
- Bo Ram Choi analyzed the data, prepared figures and/or tables, authored or reviewed drafts of the paper, approved the final draft.

### Human Ethics

The following information was supplied relating to ethical approvals (i.e., approving body and any reference numbers):

The Institutional Review Board of Seoul National University Bundang Hospital approved the study design (No. L-2016-1265).

### Data Availability

The raw data is available as a Supplemental File.

### Supplemental Information

Supplemental information for this article can be found online at http://dx.doi.org/10.7717/peerj.7275#supplemental-information.

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
