# Peer review of "Factors encouraging mobile instant messaging service use in medical education"

_PeerJ, doi:10.7717/peerj.7275_

## Round 0.1 · original submission · Major Revisions

Overall, this is an interesting article and the merits are reflected in the reviews. Three reviews were provided and overall the reviews confirm the manuscript is publishable with major revision. Toward this, I suggest working through these revisions carefully to revise this study. First, please note a need to strengthen:
1) literature review- this is limited in the paper and does not address the larger body of work in this growing area
2) methodology - strengthen and place in context of the literature (see 1 and expand) and place your statistical work in a theoretical framework. Note: the comments asking if using the virtual space is meaningful rather than causal are important to address
3) discussion revision - please see the comments requesting more detailed thought on the value of the chat and sharing rather than the amount of sharing

I look forward to a revision and encourage you to capitalize on the reviewer critiques to strengthen this interesting paper.

Reviewer 1 ·

Basic reporting

Literature review is not sufficient enough. Please see the general comments.

Experimental design

The analysis is a little superficial. Please see the general comments.

Validity of the findings

The findings themselves are fine, but the discussion does not suggest any theoretical or practical implication of the current study. Please see the general comments.

Additional comments

This study investigated the use of Kakao Talk in clinical education, by examining 39 groups of medical students’ interactive records during a one-week clinical education. It statistically analyzed the relationship between the number of posts during the first two days and the last three days. The results showed the more students posted, the more likely they would particulate academically in the discussion. Students were more likely to post academic related messages in the later half of the week. In addition, the number of posts in the first two days was positively correlated with that of the last three days.

This study deals with an interesting and relevant topic. It could be a valuable addition to the literature regarding using mobile-instant-messaging in educational settings, particularly in medical education. However, there are many issues to be addressed before the manuscript can be considered at a publishable level.
1. My biggest concern about this research is its significance. To me, the results achieved by the study is pretty obvious and self-explanatory. For instance, the more students used the service, the more likely they would share more academic information. And once they became familiar with the service, the learning environment and the content, it is predictable that students would share more academically related posts. Therefore, in addition to the analysis of the number of posts, I would personally want to see more interesting and sophisticated content analysis results. For instance, how about students’ collaborative learning, the level of thinking? Just some random thoughts.
2. Another problem I have with the manuscript is the literature review. The authors hardly reviewed any literature regarding this topic substantially, which fails to provide a solid academic context to situate the current study. Without an up-to-date and critical review of related literature, this study is hardly situated in the scholarship of this field of study.
3. What is the theoretical and practical implication of the current study? What theories did the authors refer to structure the study? After presenting the results, how would the authors interpret the achievement theoretically? Any practical suggestions?
4. I also have some questions regarding the details of the manuscript. Some are listed below as examples. The authors can address them but my biggest concerns have been stated above in points 1-3.
a) 71. More explanation is needed on “the influences on student educational performances vary”. How did it vary and why?
b) 77. What is the other media?
c) 80. Any reference for this statistic result?
d) 87. On what basis did you assume that “Kakao Talk would be used often when it is considered helpful for academic purposes?” (87)
e) 90. Where did the hypothesis that “an increase in student activity on Kakao Talk would influence its later use for academic purposes?” come from?

Reviewer 2 ·

Basic reporting

A. Clear and unambiguous, professional English used throughout.
i. Good
B. Literature references, sufficient field background/context provided.
i. Good
C. Professional article structure, figures, tables. Raw data shared.
i. Good
D. Self-contained with relevant results to hypotheses.
i. Moderate

Experimental design

A. Original primary research within Aims and Scope of the journal.
i. Good
B. Research question well defined, relevant & meaningful. It is stated how research fills an identified knowledge gap.
i. Not bad
C. Rigorous investigation performed to a high technical & ethical standard.
i. Good
D. Methods described with sufficient detail & information to replicate.
i. Good

Validity of the findings

A. Impact and novelty not assessed. Negative/inconclusive results accepted. Meaningful replication encouraged where rationale & benefit to literature is clearly stated.
i. Good
B. Data is robust, statistically sound, & controlled.
i. Good
C. Conclusions are well stated, linked to original research question & limited to supporting results.
i. Moderate
D. Speculation is welcome, but should be identified as such.
i. Good

Additional comments

A. Overall
i. This study is not a confirmation of causality (“Kakao talk might be useful to medical students’ educations”), but a relevance problem (“MIM service use during the early period of clinical education might related to use for academic
ii. purposes during later periods.”). Therefore, causal expressions throughout the thesis must be replaced with relevance expressions in whole manuscript.
iii. The reference format of the introduction section (line 71; 1-9) and discussion section (line 200; [7]) is different.

B. Abstract
i. Line 49; Please replace “influence” to “relation”

C. Introduction
i. Line 80; spelling error (aka Ka Talk, Kakao Corp)
ii. Line 86; Please, remove the following hypothesis.
1. “Moreover, there might be factors that encourage the use of Kakao Talk for academic purposes and increase students’ familiarity with its educational use.”
iii. Line 86; Please, remove the following hypothesis.
1. “We hypothesized that an increase in student activity on Kakao Talk would influence its later use for academic purposes. Therefore, this study investigated whether, in our hospital, student activity on Kakao Talk early in the clinical education period influenced Kakao Talk activities for academic purposes later in the clinical period.”

D. Discussion
i. Line 216~219; I understand that early Kakao talk usage is related to late Kakao talks usage for academic purposes. Why does this result lead to Kakao talk boosting students' educational outcome? It can be interpreted that students with a tendency to use messenger in early period more often use messenger for academic purposes in late period. To assert that Kakao talk has increased students' educational outcome, you should compare the academic achievement between the groups with and without Kakao talk. Therefore, the conclusion should be changed as follows. “Early Kakao talk usage is related to late Kakao talks usage for academic purposes.”

Reviewer 3 ·

Basic reporting

The article must be expressed clearly in English.
1. Abstract should be arranged clearly. Introduction needs to be added with solid grounds on why this study can be explored by using references.
2. Mehtods also should be written simply to make readers understand clearly. The extended sentences listing all variables can make the article confused.
3. The structure of the article should be rearranged. In particular, in Methods, subheadings are needed.
4. Reference should conform to format.

Experimental design

1. Study design: to make it clear, Figure on the process of the experiment can be helpful.
2. The number of participant including students and teachers (professors and fellows) can be written as (n= ), which can make the extended sentences simple and clear.
3. Research questions should be added.
4. The data were analyzed rigorously. However, the article emphasizes just the statistical findings (number) from the collected data. The statement on the detailed information which could cause students' chats on virtual space should be presented.

Validity of the findings

The discussion should be appropriately stated as some parts don't show the consistency of the results. The article can be better if dividing the section into 2 parts, Discussion and conclusion

Additional comments

I appreciate your raw data and effort
However, overall structure and statement should be modified.

Annotated reviews are not available for download in order to protect the identity of reviewers who chose to remain anonymous.

---

## Round 0.2 · accepted · Accept

Thank you for a diligent and detailed effort to overhaul this manuscript. As stated in the initial review, the study merited publication with a reworking of the manuscript. That was completed in this draft. Thank you.